# Discrete Element Method Evaluation of Triboelectric Charging Due to Powder Handling in the Capsule of a DPI

**DOI:** 10.3390/pharmaceutics15061762

**Published:** 2023-06-18

**Authors:** Francesca Orsola Alfano, Alberto Di Renzo, Francesco Paolo Di Maio

**Affiliations:** DIMES Department, University of Calabria, 87036 Rende, Italy; francesco.dimaio@unical.it

**Keywords:** triboelectric charging, electrostatics, DPI, DEM, inhalation, carrier-based formulation

## Abstract

The generation and accumulation of an electrostatic charge from handling pharmaceutical powders is a well-known phenomenon, given the insulating nature of most APIs (Active Pharmaceutical Ingredients) and excipients. In capsule-based DPIs (Dry Powder Inhalers), the formulation is stored in a gelatine capsule placed in the inhaler just before inhalation. The action of capsule filling, as well as tumbling or vibration effects during the capsule life cycle, implies a consistent amount of particle–particle and particle–wall contacts. A significant contact-induced electrostatic charging can then take place, potentially affecting the inhaler’s efficiency. DEM (Discrete Element Method) simulations were performed on a carrier-based DPI formulation (salbutamol–lactose) to evaluate such effects. After performing a comparison with the experimental data on a carrier-only system under similar conditions, a detailed analysis was conducted on two carrier–API configurations with different API loadings per carrier particle. The charge acquired by the two solid phases was tracked in both the initial particle settling and the capsule shaking process. Alternating positive–negative charging was observed. Particle charging was then investigated in relation to the collision statistics, tracking the particle–particle and particle–wall events for the carrier and API. Finally, an analysis of the relative importance of electrostatic, cohesive/adhesive, and inertial forces allowed the importance of each term in determining the trajectory of the powder particles to be estimated.

## 1. Introduction

Pharmaceutical powders are used in various stages of drug manufacturing and are an essential part of the pharmaceutical industry. As for most non-conductive materials, the handling and processing of these powders often leads to the transfer and accumulation of electrostatic charges. This, in turn, can cause issues in drug manufacturing, such as cross-contamination, intermittent powder flow, and even explosion hazards [1]. Poor powder flow can lead to difficulties in powder handling, filling, and packaging processes [2]. The agglomeration of particles can affect the powder’s dissolution rate, which can impact the drug’s bioavailability [3]. Segregation of the powder blend can result in non-uniformity of the drug content and, consequently, inconsistent dosing [4]. Therefore, understanding the mechanisms of electrostatic charge transfer and accumulation in pharmaceutical powders is crucial for ensuring the safety and quality of drug manufacturing processes.

Tribocharging is the process responsible for the charge transfer between contacting material surfaces, and it takes place upon detachment [5]. Carrier-based inhalation formulations used in Dry Powder Inhalers (DPIs) are composed of two solid components (i.e., the active pharmaceutical ingredient (API) and the carrier) with significantly different sizes and exhibit a complex tribocharging behavior. Effects such as the bipolar charging phenomena, i.e., particles the same material charged both positively and negatively, have been observed [6]. In some cases, experimental studies that focused on the tribocharging phenomenon [7,8,9,10,11] reported somewhat conflicting results, even in the polarity of the reported charge, calling for additional investigations at small scales. The charge polarity of lactose particles (a widely used carrier and excipient) depends on the chosen inhaler [8], on the material of the containing capsule [12], and even on their manufacturing process; for example, milled lactose tends to charge positively, while sieved lactose tends to charge negatively [8]. Salbutamol sulphate (a commonly used API in inhalation formulations) tends to acquire a positive charge if it is amorphous and a negative charge if it is in its crystalline form [11]. Peart [13] reports that salbutamol charges positively when in contact with PVC, while lactose charges negatively. However, when salbutamol particles detach from lactose, the opposite polarity is measured for the two materials (salbutamol’s specific charge is about −3000 nC/g, and lactose’s specific charge is about 100 nC/g).

When strong size polydispersity is present, the charge polarity also depends on particle size, with smaller particles that charge negatively and larger particles that charge positively [11,14,15,16]. This effect cannot be predicted using the triboelectric series, which is usually employed to rank the electronegativity of powder materials [1,2]. The concentration of the API plays a role as well, with the electrostatic charge of drug–carrier mixtures usually decreasing by increasing the concentration of the API [10,17,18].

Electrostatics also plays a major role in the design and use of hard capsules for capsule-based DPI. Chow et al. [19] found that mechanical vibration such as tapping induced significant static charge on lactose stored in a gelatine capsule. Hoe et al. [10,17] hypothesized that the surface charge on the capsule might be high enough to ionize the surrounding air. Understanding such a variety of observations requires a careful analysis of the links between microscopic charge transfer processes and the macroscopic manifestations. Particle-scale information is an important ingredient that is accessible only through simulation.

In recent years, Discrete Element Method (DEM) modeling has emerged as a powerful simulation tool for studying the behavior of powders in inhalation devices [20]. Coupled DEM-CFD (Computational Fluid Dynamics) studies showed the detailed motion of particles from the initial dispersion in air through to the mouthpiece [21,22,23,24,25]. In the last decade, model formulations have been introduced to extend the particle-scale contact tracking capability of DEM with surface–surface charge transfer and physical electrostatic interaction models (see, e.g., [26,27,28]), allowing triboelectric charging phenomena at the particle scale to be studied [27,29,30,31,32,33]. Naik et al. [34] studied the triboelectrification of binary mixtures of drug and excipient in a blender. They found that particle–particle interactions enhance the electrostatic interaction between the drug and excipient and decrease the overall charge transfer between particles and walls. The overall charging process that resulted was mitigated by this effect, leading to a lower total charge than that acquired by single components, and differences in excipient concentration, in some cases, caused charge polarity reversal. The importance of particle–particle contact charging was also highlighted by Chowdhury et al. [33]. Zhu et al. [35] studied the contact electrification effect of selected API agglomerates in the Turbuhaler^®^, finding a reduction in inhaler efficiency due to the triboelectrification of powders. Specific numerical studies on the motion of particles in the capsule of capsule-based DPIs are available in the literature [36,37,38,39], but the effects of charges generated upon contact on the release of powder from the capsule itself are not taken into account. Most of previous works in DPI applications focus on the influence of electrostatic interactions between previously charged particles, rather than the charge buildup process, and the charging of capsule walls is typically neglected.

The aim of the present study is to apply extended DEM simulations to evaluate triboelectric charging dynamics of a lactose–salbutamol binary mixture in the chargeable capsule of a DPI. The capsule is vibrated to simulate the routine handling operations to which a filled capsule is subjected during its life cycle, with the aim to assess the extent to which such movements give rise to electrostatic charge accumulation. The simulation tool offers the possibility to relate individual contact events and local charge transfer to the macroscopic influence that such a charge exerts on the material dynamics. The model formulation, simulation setup, and material parameter are presented first. After validation, the results of simulations of representative systems at low- and high-dosage conditions are discussed.

## 2. Materials and Methods

### 2.1. DEM Simulation Technique

DEM is a numerical method for simulating the behavior of a system of discrete particles by tracking their individual motions and interactions due to the forces and momenta acting on each of them. Spherical particles are considered in the present work.

The translational (${\vec{v}}_{i}$) and rotational (${\vec{\omega }}_{i}$) velocities of the *i*-th particle are calculated by integrating Newton’s second law of motion:(1)$$m_{i}\frac{d{\vec{v}}_{i}}{dt}={\vec{F}}_{T}=\overset{Nc}{\underset{j=1}{\sum }}{\vec{F}}_{i,j}^{c}+\overset{Nc}{\underset{j=1}{\sum }}{\vec{F}}_{i,j}^{e}+{\vec{F}}_{g,i}$$
(2)$$I_{i}\frac{d{\vec{\omega }}_{i}}{dt}={\vec{T}}_{T}=\overset{Nc}{\underset{j=1}{\sum }}{\vec{T}}_{i,j}^{c}+{\vec{T}}_{r}$$
where ${\vec{F}}_{T}$ is the sum of all the forces acting in the *i*-th particle—contact, electrostatic, and gravitational forces; and ${\vec{T}}_{T}$ is the sum of all the torques acting on the *i*-th particle—contact torque and rolling resistance, $T_{r}$.

Cohesive-contact forces are modeled following the JKR contact theory [40], which accounts for attractive forces due to van der Waals effects and is hysteretic, i.e., loading and unloading cases are different in that cohesive contact is initiated at zero distance, while detachment occurs at non-zero distance. A velocity-dependent dissipation is also introduced to be able to model (coefficient of) restitution after impacts. The magnitude of the normal contact force between two contacting particles is given by the following:(3)$$F_{c,ij}^{n}=4\sqrt{\pi \gamma E_{eq}}a^{\frac{3}{2}}-\frac{4E_{eq}}{3R_{eq}}a^{3}-\eta_{n}^{H}\delta_{n}^{\frac{1}{4}}v_{n}$$
where γ is the surface energy, a is the radius of the contact area, *v_n_* is the normal velocity, $\delta_{n}$ is the normal overlap, $\eta_{n}^{H}$ is the normal damping coefficient (related to the restitution coefficient; see, e.g., [41]), and $E_{eq}$ and $R_{eq}$ are the equivalent Young modulus and radius of the two contacting particles (*i* and *j*) [42].

The normal overlap, $\delta_{n}$, is related to the radius of the contact area as follows:(4)$$\delta_{n}=\frac{a^{2}}{R_{eq}}-\sqrt{\frac{4\pi \gamma a}{E_{eq}}}$$

As mentioned above, according to the JKR theory, during the detachment phase, the contact remains active at negative overlaps between the spheres (as actual surfaces are elongated shapes that are still in contact) until a threshold overlap is reached. The maximum attractive force, usually referred to as the pull-off force, occurs at a negative overlap and is given by the following:(5)$$F_{pull-off}=3\pi \gamma R_{eq}$$

The tangential contribution to the contact is considered following the no-slip solution of Mindlin and Deresiewicz [43] for the frictional–elastic part and a velocity-dependent dissipation term similar to the normal direction. The tangential contact force is calculated as follows:(6)$$F_{c,ij}^{t}=-\left(\mu_{s}F_{c,ij}^{n},\;8G_{eq}\sqrt{R_{eq}}\delta_{n}^{\frac{1}{2}}\delta_{t}+\eta_{t}^{H}\delta_{n}^{\frac{1}{4}}v_{t}\right)\;$$
where *μ_s_* is the static friction coefficient, $G_{eq}$ is the equivalent shear modulus, $\delta_{t}$ is the tangential overlap, $\eta_{t}^{H}$ is the tangential damping coefficient, and $v_{t}$ is the tangential velocity.

In the rotational motion, the contact torque results from the action of the tangential contact force. The rolling friction torque is calculated according to the Constant Directional Torque model [44], introducing the rolling friction coefficient, $\mu_{r}$, as material parameter.

All the models presented above are described in more detail in Alfano et al. [39,45].

The DEM simulations were carried out using an in-house customized version of the open-source code MFIX (NETL MFS, Department of Energy (Morgantown, WV, USA), version 18.1.5 [46]. Johnson-Kendall-Roberts (JKR) model for the cohesive force and constant directional torque (CDT) model for the rolling friction were implemented in the original version of the code (see [45] for more details). Moreover, a different approach for wall contacts [47] was preferred to the standard MFIX formulation. Special precautions have also been taken to prevent very fine particles from being unrealistically pushed out of the domain by the carrier particles, crossing the domain boundary. In terms of hardware resources, an own-managed cluster was extensively utilized by running parallel tests on up to 32 cores.

### 2.2. Triboelectric Charging Model and Electrostatic Forces

Consideration of the charge transfer, buildup, and corresponding modification to the powder dynamics requires two essential ingredients: a tribocharging model for the charge transfer between surfaces after contact and a model for the interaction force between charged bodies. Our selection for both models is introduced below.

The triboelectric charging model is based on the condenser model developed by Matsusaka et al. [48] in the formulation for DEM implementation proposed by Pei et al. [27]. It was extensively presented in a previous work [31] and is shortly summarized here.

Each particle and wall element is assigned a net scalar charge that is representative of its surface-charge distribution. This charge evolves as a result of particle–particle and particle–wall collisions and contacts (Figure 1a) upon surface detachment. The charge transferred after each particle–wall impact is calculated as follows:(7)$$\Delta q=kS_{m}\left(\frac{\Phi_{i}-\Phi_{s}}{e}+\xi \frac{z_{s}}{4\pi \varepsilon_{0}}\frac{q_{i}}{R_{i}^{2}}\right)$$
where $\Phi_{i}$ and $\Phi_{s}$ are the work functions of the particle and the wall surface, respectively; e is the elementary charge (1.602 × 10^−19^ C); $R_{i}$ is the particle radius; $z_{s}$ is the cutoff distance for particle–wall charge transfer (considered as 130 nm [27]); $q_{i}$ is the charge on the particle before impact; ξ is the image correction factor [49], which is set to 2; $\varepsilon_{0}$ is the vacuum permittivity (8.854 pF/m); *k* is the charge efficiency, which is set to 10^−4^ C m^−2^ V^−1^ [27]; and $S_{m}$ is the maximum contact area, which is calculated as follows:(8)$$S_{m}=\pi \delta_{n,max}R_{eq}$$
where $\delta_{n,max}$ is the maximum normal overlap and $R_{eq}$ is the equivalent radius (harmonic mean) between the two contacting surfaces.

The first term of the sum in brackets in Equation (7) is the contact potential difference, i.e., the driving force for charge exchange, while the second term arises from the image effect. Note that charge transfer occurs, increasing one surface by some charge and decreasing the other one by the same amount, so that the total system charge remains constant.

The particle–particle charge transfer is calculated in a similar fashion [27]:(9)$$\Delta q=kS_{m}\left(\frac{\Phi_{i}-\Phi_{j}}{e}+\xi \frac{z_{p}}{4\pi \varepsilon_{0}}\left(\frac{q_{i}}{R_{j}^{2}}-\frac{q_{j}}{R_{i}^{2}}\right)\right)$$
where $z_{p}$ is the cutoff distance for the particle–particle charge transfer, and it is set to 260 nm [27].

Once each particle has its own charge, the second ingredient of the model, the electrostatic force between two charged particles (Figure 1b), is calculated according to Coulomb’s law:(10)$${\vec{F}}_{i,j}^{e}=\frac{1}{4\pi \varepsilon_{0}}\frac{q_{i}q_{j}}{r_{i,j}^{2}}{\hat{n}}_{i,j}$$
where ${\hat{n}}_{i,j}$ is the unit vector defined along the direction connecting the two particles’ centers, and $r_{i,j}$ is the distance between these centers. The cutoff distance for the calculation of the electrostatic interactions is set to 1.2 times the sum of the two particles’ radii.

Coulombic interactions between charged particles and the walls of the capsule were considered according to the method of mirror charges [50]:(11)$${\vec{F}}_{i,s}^{e}=\frac{1}{4\pi \varepsilon_{0}}\frac{q_{i}^{2}}{{\left(2r_{i,s}\right)}^{2}}{\hat{n}}_{i,s}$$
where $r_{i,s}$ is the distance between the wall surface and the center of the particle, and ${\hat{n}}_{i,s}$ is the unit vector perpendicular to the surface and passing through the particle center. It is useful to note that while particle–particle interactions may be attractive or repulsive depending on the charge polarity, particle–wall electrostatic interactions are always attractive.

### 2.3. Simulation Parameters

The reference materials for API and carrier particles are salbutamol and lactose, respectively. The diameter of the salbutamol particles is set to 5 μm, while the diameter of the lactose particles is 100 μm, considering the commercially available Inhalac^®^ 230 as a reference.

The DEM simulation properties are reported in Table 1. The mechanical data are selected according to typical values found in the literature [51,52].

The work functions were calculated from molecular orbital calculation (MOPAC) by Naik et al. [9] for the lactose and by Zellnitz et al. [53] for salbutamol. The work function of the gelatine capsule was not found in the literature. Pinto et al. [12] observed that the charge acquired by the capsules in contact with stainless steel is about 60% of the charge acquired by the capsules in contact with PVC. Since the work function of PVC is 5.33 eV [34] and that of stainless steel is 5.05 eV [54], and since the charge is proportional to the difference between work functions (Equation (9)), the value Φ= 4.60 eV was estimated.

By looking at the work functions, salbutamol is expected to become negatively charged, while the capsule is expected to become positively charged. The behavior of the carrier particles is less predictable: lactose will acquire a positive charge after contact with the API and a negative charge if in contact with the capsule.

The cohesion properties are reported in Table 2 for each material pair [52,55].

### 2.4. Geometry and Powder Configurations

The geometry used in the simulations is modeled using a size 3 hard-shell gelatine capsule [56] as a reference. The capsule is filled with a carrier–API blend. Two powder configurations are considered, as described in Table 3 and shown in Figure 2.

Configuration A (Figure 2a) has a total mass of 25 mg. It consists of about 31k carrier particles and 93k API particles. Configuration B (Figure 2b) has a total mass of 1 mg and is made of about 1500 carrier particles and 117k of API particles. The total number of particles is similar, but the mixing ratio between the two solid phases differs consistently: 1:3332 (0.03%, *w/w*) for Configuration A, and 1:124 (0.80%, *w/w*) for Configuration B.

The approach for the dry powder coating of the carrier with API is described in Alfano et al. [22]. The difference in the coating degree between Configuration A and B can be noticed in Figure 3, which shows two coated carrier particles.

The coated carrier particles are initially placed in a regular cubic arrangement, which was visualized in Figure 2. The initial random condition is obtained by letting them settle under gravity. Then, the capsule is subject to a periodic oscillatory translation to reproduce a tapping motion. The shaking frequency is 120 Hz, and the shaking amplitude is 0.8 mm. The shaking direction changes during the simulations, so that the capsule shakes alternatively in both horizontal and vertical directions (x and y, respectively). In implementation, rather than computing the actual motion of the capsule, the motion is tracked in the frame of reference of the capsule by adding the equivalent corresponding fictitious forces on the particles. More details of this implementation in MFIX can be found in Alfano et al. [39].

## 3. Results and Discussion

### 3.1. Model Validation

To verify the reliability of the tribocharging model with the selected parameters, a comparison was made with the data reported by Chow et al. [19], who studied the effect of powder handling by repeatedly tapping a gelatine capsule filled with 25 mg of lactose (InhaLac^®^ 230, d_50_ = 97.2 μm). The specific charge of the sample was measured after 0, 10, 40, 100, and 200 taps.

To reproduce the tapping, a shaking motion with a 120 Hz frequency and 0.8 mm amplitude was considered. The simulation time was set to 1.67 s in order to have 200 full oscillations, corresponding to 200 taps. The capsule was filled with 25 mg of 100 μm lactose particles, initially settled at the bottom of the capsule. No API was included in this case. The initial charge of the particles was set according to a normal distribution with a mean of −0.025 pC (corresponding to the initial specific charge in the reference article, −0.779 nC/g, multiplied by the weight of a single particle) and standard deviation of 25% times the mean. The initial charge distribution is shown in Figure 4a.

Figure 4b shows the total charge of the sample as a function of the number of taps, which also corresponds to the time evolution. The solid line is the result of the simulation, and the dots are the reference experimental data [19]. The increase in negative total charge follows the trend exhibited by the experimental measurements, well within the experimental uncertainty, except for the second point. Indeed, the data by Chow et al. [19] show a sudden jump in the charge, followed by a somewhat stationary stage between 10 and 40 taps. The simulations show a more gradual increase in the acquired charge, which may be helped by the ability to track the charge at every instant during the simulation. Attempts have been made to better reproduce the charge corresponding to the second point by considering tri-disperse particle sizes (10%vol 60 μm, 80%vol 100 μm, and 10%vol 150 μm) by changing the capsule wall’s work function and the charging efficiency coefficient. In no case, however, did the abrupt initial change and subsequent less pronounced increase turn out to be predictable. Overall, the agreement can be judged to be sufficiently good to proceed with the more complex carrier–API cases.

### 3.2. Gravity Settling in the Capsule

Starting from the ordered configurations of the powder shown in Figure 2, with both carrier and API particles, DEM gravity settling simulations have been performed to evaluate the charge buildup due to operations such as capsule filling and storage. Figure 5 shows the final configuration after the settling simulations. Figure 5a shows Configuration A (25 mg) after 40 ms, with the powder bed stable at the bottom of the capsule. A much longer simulation was required for Configuration B (1 mg), as after 40 ms, the particles were still consistently bouncing off the walls of the capsule. The simulation was extended up to 200 ms when most of the powder bed was settled (Figure 5b), and the charge level reached a stationary point (see below).

Figure 6 shows the total charge acquired by the samples following the deposition simulations. The API and carrier curves are shown individually, as well as the total charge curve. Note that the charges are evaluated at very short time increments, and the symbols on the plot lines are only for reference. Contrary to the simulation with only lactose particles (Figure 4b), which became negatively charged, in this case, the carrier acquires a positive charge, while the salbutamol acquires a negative charge. Figure 6a shows some oscillations in the total charge of the carrier: first, it increases, then it decreases, and then it increases again, probably depending on the instantaneous ratio between carrier–wall and carrier–API interactions. Similar fluctuations can be observed in Figure 6b, which shows the result with the 1 mg sample and a different mixing ratio, but in a less pronounced way, suggesting that, in this case, the greater quantity of active principle and the reduced number of carrier particles promote API–carrier interactions at the expense of carrier–wall interactions. In both cases, the total net charge is negative and has a greater magnitude for the larger sample. A plateau in the contact charging process is observed more markedly in the 25 mg case.

In Figure 7, the specific charge evolutions (expressed as the charge-to-surface ratio, CTS) in the first 40 ms of the two simulations for carrier and API particles are reported. Carrier particles in the 1 mg simulation acquire a higher specific charge than carrier particles in the 25 mg case. The opposite behavior is observed for API particles, for which the specific charge magnitude is higher for the 25 mg simulation.

In Table 4, a summary of the net and specific charges after gravity settling is reported. The 25 mg test shows a higher net charge (four times higher), but the specific charge is higher for the 1 mg test, whether it is expressed as the CTS (charge-to-surface ratio) or as the CTM (charge-to-mass ratio). The numerical value of the CTS is equal to the CTM for API particles, since the mass/surface ratio is 1 g/m^2^.

Compared to the already mentioned specific charge of 25 mg lactose powder stored overnight in a gelatine capsule reported by Chow et al. [19], CTM = −0.779 nC/g, a lower value is obtained with the simulations (−0.38 nC/g), suggesting that the presence of salbutamol as API mitigates the powder charging.

### 3.3. Tapping of the Capsule

The capsule was subjected to the vibrating motion presented in Section 3.1. in the presence of both API and carrier particles. The configuration obtained after gravity settling (Figure 5) was taken as the initial condition, considering the charge recorded after gravity settling. The simulation was carried out for 400 ms, corresponding to 48 taps, with a shaking frequency of 120 Hz.

Figure 8 shows the total net charge acquired by the 25 mg (Figure 8a) and 1 mg (Figure 8b) samples after capsule shaking. Now, the lactose carrier particles acquire an overall negative charge, as in the simulations with carrier particles only (Figure 4b). The total net charge is negative in both the 1 mg and 25 mg samples. The final charge is about −0.6 nC for the 1 mg sample and −0.8 nC for the 25 mg sample. The charge acquired by 25 mg of carrier particles (Figure 4b) shaken 50 times was more than twice, suggesting again the possible mitigatory effect [9,10] in the total charge acquisition due to the presence of salbutamol API particles.

The evolution of the specific charge is reported in Figure 9 as the charge-to-mass ratio (CTM) and in Figure 10 as the charge-to-surface ratio (CTS). The specific charge recorded for the API particles (Figure 9a and Figure 10a) is substantial, exceeding −50,000 nC/g (or −50,000 nC/m^2^) with both capsule loadings. The curves for the 25 mg case show a somewhat flattening trend at the end of the simulations, suggesting the beginning of a plateau phase.

Observing the evolution of carrier particles’ specific charge (Figure 9b and Figure 10b), the curve for the 1 mg sample is of particular interest. An initial positive charge buildup is observed, followed by a decrease in the charge and a subsequent markedly linear trend, until it reaches a specific charge value of almost −130 nC/g (or −3000 nC/m^2^). The specific charge for the 25 mg sample is more than one order of magnitude lower.

Figure 11 shows the individual charge distribution for API (top) and carrier (bottom) particles at different times: 50 ms, 200 ms, and 400 ms. The charge distribution of API particles becomes wider, going from 50 ms to 200 ms, slightly narrowing again at the end of the simulation (Figure 11c, top). The mean value shifts to the left, i.e., with an increase in charge with negative polarity.

The evolution of carrier particles’ charge distribution is different for the 1 mg and 25 mg samples. With the 25 mg sample, more and more carrier particles acquire a positive charge over time, while the opposite tendency is observed for the 1 mg sample: after 50 ms, most of the particles carry a positive charge (Figure 11a, bottom), while no carrier particle is charged positively at the end of the simulation (Figure 11c, bottom).

Figure 12 shows a colored map of the surface charge density (σ) observed in the capsule after shaking in the *x*-direction for 60 ms, as estimated according to Equation (9). The charge is positive, as the wall has the lowest work function. Comparable charge density values are observed between the two different loadings. However, a wider and more scattered area is associated to the 1 mg sample case, probably due to the more chaotic movement of API particles, while the 25 mg case is characterized by higher values localized near the main impact locations (normal to the shaking direction). The charge density estimated for the capsule is about 10 times lower than the CTS of carrier particles in the 1 mg configuration (see Figure 10b), while the opposite trend is observed in the 25 mg simulation. On the other hand, in both configurations, the CTS calculated for API particles is significantly higher than the capsule surface charge density (see Figure 10a).

### 3.4. Collision Statistics

To interpret the charge acquisition data, it is useful to relate the evolution of the acquired charge to the events that generate the charge transfer, i.e., the collisions. To that purpose, the MFIX code was modified to track individual collision events in DEM separately for particle–particle and particle–wall cases.

Figure 13a shows the percentage of API particles in contact with the walls of the capsule as a function of time. In terms of the fraction of the total, the 1 mg case shows a higher value, but the time variation is not significant in both cases. Figure 13b shows the evolution of the carrier-to-API coordination number (CN), i.e., the number of API particles in contact with a carrier particle averaged over all carrier particles. Interestingly, the CN remains almost constant in the carrier-rich 25 mg simulation, while in the API-rich 1 mg simulation, the carrier CN goes from about 100 to 4 within the first 100 ms, meaning that collisions are strong enough to determine that detachment of most of the API particles from the carrier. This can be the explanation of the shift in the carrier charge distribution, in which the mean value goes from a positive value at 0.05 s (Figure 11a, bottom) to a progressively negative value (Figure 11b,c, bottom), as the carrier–API contact is the event that can give rise to a positively charged carrier particle.

Figure 14a shows the instantaneous number of collision events during the last part of the simulation (from 300 to 400 ms); the data for particle–particle (PP) and particle–wall (PW) contacts are presented separately. In the 25 mg simulation, more than 10^4^ particle–particle contacts are recorded, while the number of particle–wall collision events does not exceed 1000. The contact events in the 1 mg simulation stand in between, with more particle–wall contacts than particle–particle contacts, most probably due to the lower total solid mass. Figure 14b shows the average normal impact velocity; in this case, the velocity values are lower for the 25 mg simulations. Particle–wall contacts tend to occur with a higher impact velocity in both configurations.

The contact statistics data are completed in Figure 15, which shows how the collision events are distributed among the various phases involved. The majority of the collisions in the 25 mg simulations are low-energy carrier–carrier collisions, during which the charge exchanged is minimal since particles are made of the same material. The large number of carrier–carrier collisions could also be the reason why settling occurs much faster than in the 1 mg simulation, since all of these collisions dissipate the initial kinetic energy of the particles.

### 3.5. Estimate of Force Contributions

Given the significant value of the observed electrostatic charges (see, e.g., Figure 8), it is interesting to investigate whether or not the consequent electrostatic forces actually influence particle trajectories and to what extent.

By equating the weight force of the particle with the particle–wall electrostatic force calculated according to Equation (11), it is possible to estimate the charge magnitude necessary for the particle to remain attached to the wall, assuming for simplicity that the electrostatic attractive force acts in the opposite direction to the gravitational force:(12)$$\left|q_{W}\right|=16\pi R^{2}\sqrt{\frac{\varepsilon_{0}\rho Rg}{3}\;}$$

Figure 16 shows $\left|q_{W}\right|$ as a function of particle diameter. The final charge magnitude distribution is also reported in the form of a scatter plot for the two systems investigated. Most API particles at the end of the simulation possess a charge level that is higher than $\left|q_{W}\right|$, while no carrier particle exceeds such a value (all points are below the blue curve). The results suggest that during capsule handling and shaking, a fraction of API particles are likely to end up retained inside the capsule, on its internal walls, due to the charge buildup, while this is much less likely to happen for a carrier particle.

It is useful to note that other cohesive/adhesive forces and inertial fictitious forces are at play (see, e.g., Table 2), so their relative contribution should be compared. An evaluation in terms of the maximum force values is discussed below.

The maximum electrostatic force was calculated with the Coulomb law, considering the maximum particle charge magnitude in both simulations, which is 7.7 fC for API particles and 170 fC for carrier particles. For the calculation of the distance of the Coulomb force, it is assumed that the two bodies are in contact; therefore, for example, in the case of a p-P-P interaction, the sum of the particle radii was considered.

To estimate the maximum cohesive and adhesive forces, the reference values of the pull-off force in the JKR model are considered, i.e., the values reported in Table 2.

The fictitious force associated with the shaking motion follows a sinusoidal trend in time. Its maximum value is given by the following:(13)$$F_{shk}=4\pi^{2}Af^{2}\times m_{p}$$
where *A* is the shaking amplitude, *f* is the shaking frequency, and *m_p_* is the mass of the particle. In the present study, this force reaches about 50 times the weight of the particle.

Figure 17 shows a comparison between the estimated maximum force contribution for carrier and API particles. As expected from the externally imposed motion, the fictitious force associated with shaking primarily determines the motion of the carrier particles, as it is higher than any other contributions. Electrostatic forces are lower than cohesive forces, with the API–carrier electrostatic interaction being the lowest (and lower than weight). On the other hand, the adhesive API–carrier force contribution is dominant in the case of API particles, the fictitious force appears to be irrelevant compared to the other forces at play, and the electrostatic forces are relevant.

It is interesting to note that the electrostatic API–API force is higher than the cohesive JKR reference force, suggesting that the repulsive Coulomb force between like-charged particles can overcome the attractive van der Waals interactions. In reality, with sufficient charge difference, even like-charged particles can be attracted to one another due to a mutual polarization of surface charge [57,58]. This is not accounted for in simple Coulombic interactions, as modeled here, and more sophisticated approaches would be necessary.

## 4. Conclusions

An extended DEM approach was applied to evaluate triboelectric charging of a lactose–salbutamol binary mixture in the gelatine capsule of a DPI. Two powder configurations were considered, with different loaded doses and mixing ratios. The selected triboelectric charging model and parameters were validated with the experimental data available in the literature. Unlike in previous works, dynamic charging and interparticle electrostatic interactions of carrier and API powders were detailed investigated, including the charge buildup on the capsule walls. The detailed analysis of the flow and collision behavior during tapping allowed us to elucidate the mechanisms leading to the final charge polarity of all three materials, a condition not predictable a priori.

Gravity settling simulations were performed to estimate the charge buildup due to operations such as capsule filling and storage. The total net charge after settling was found to be negative, despite the opposite charge polarity acquired by the carrier (positive) and API (negative). Then, the capsule was subjected to a vibrating motion to simulate routine handling operations and check whether such movements can give rise to a significant electrostatic charge. A consistently higher charge buildup was measured with the vibrating motion compared to the gravity settling simulations, with an overall negative charge also for carrier particles. The net charge was higher for the carrier-rich 25 mg formulation, but the specific charge was higher for the API-rich 1 mg formulation. The two different configurations also show a different charge distribution, with a more pronounced bipolar charging tendency in the case of the 25 mg dose. To interpret the charging dynamics, a detailed study of contact statistics was performed. The mechanical shaking movement promoted API detachment from the carrier, especially in the 1 mg configuration (with higher API dosage). With the 25 mg configuration, a high number of low-energy carrier-carrier collisions were recorded which tended to dissipate the kinetic energy of the particles. On the other hand, API–API and API–wall collisions were prevalent in the 1 mg configuration. Finally, a comparison of the forces at play revealed that electrostatic forces are relevant for API particles and might play a major role in cohesion and adhesion phenomena. The present work lays the foundation for new developments in the relations between particle-scale charge transfer and buildup and the macroscopic manifestations, particularly in relation to DPI performances.

## Figures and Tables

**Figure 1 pharmaceutics-15-01762-f001:**
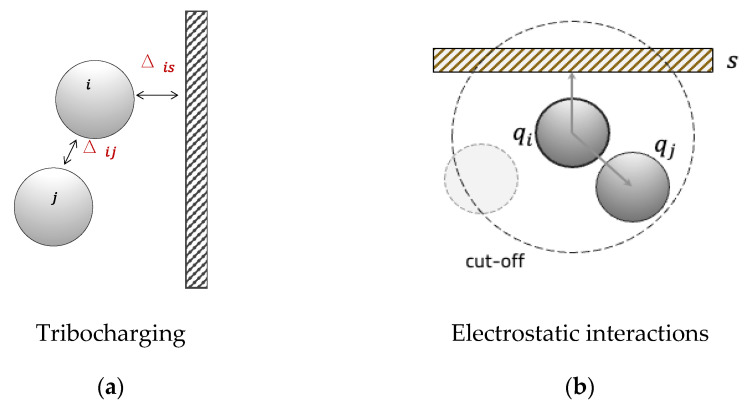
(**a**) Schematics of contact charging mechanism. (**b**) Graphical representation of particle–particle electrostatic forces evaluated within a cutoff distance.

**Figure 2 pharmaceutics-15-01762-f002:**
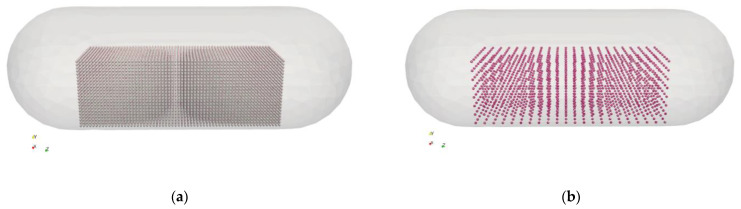
Initial location of powder with the two powder loadings: (**a**) 25 mg and (**b**) 1 mg.

**Figure 3 pharmaceutics-15-01762-f003:**
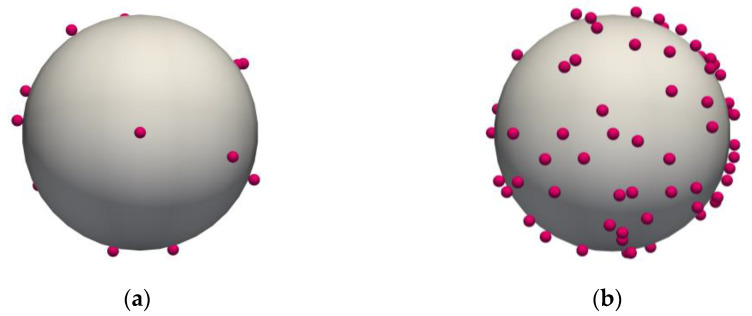
API-coated carrier particles in the two configurations considered: (**a**) 25 mg, 1:3332 (*w/w*); and (**b**) 1 mg, 1:124 (*w/w*).

**Figure 4 pharmaceutics-15-01762-f004:**
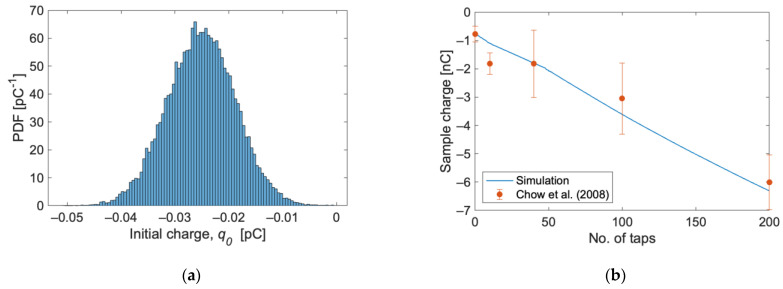
(**a**) Initial charge distribution and (**b**) sample charge as a function of the number of taps. The line is the result of the simulations, and the points are experimental data reported by Chow et al. [19].

**Figure 5 pharmaceutics-15-01762-f005:**
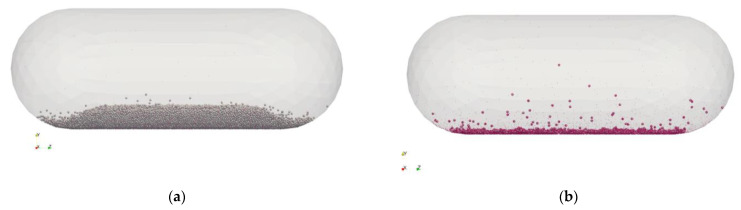
Powder configuration after gravity settling: (**a**) 25 mg (after 40 ms) and (**b**) 1 mg (after 200 ms). Carrier particles are shown in gray, and API particles are in purple.

**Figure 6 pharmaceutics-15-01762-f006:**
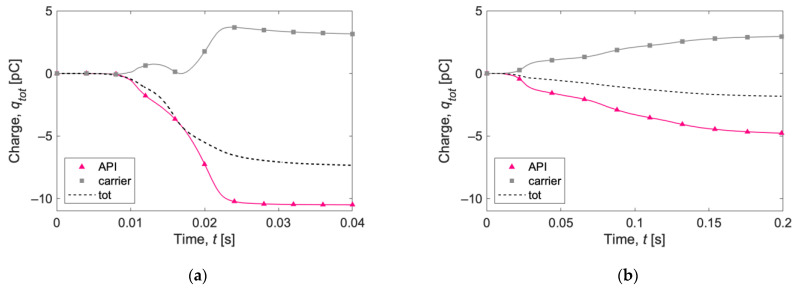
Total charge acquired by (**a**) 25 mg and (**b**) 1 mg of powder after gravity settling in the capsule. The lines show instantaneous evaluation of the charges, and the symbols are only for references.

**Figure 7 pharmaceutics-15-01762-f007:**
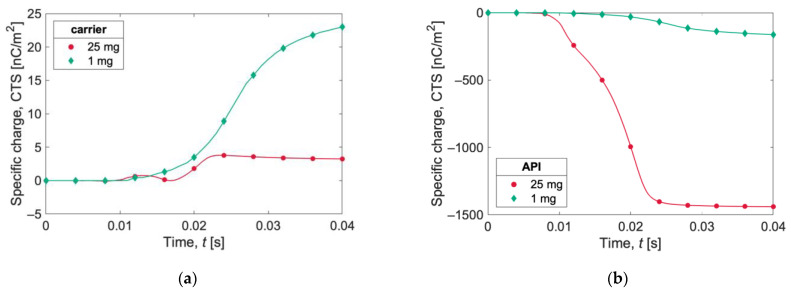
Charge-to-surface ratio (CTS) acquired during gravity settling by (**a**) carrier and (**b**) API for the two simulations (1 mg and 25 mg). The lines show the instantaneous evaluation of the charges, and the symbols are only for references.

**Figure 8 pharmaceutics-15-01762-f008:**
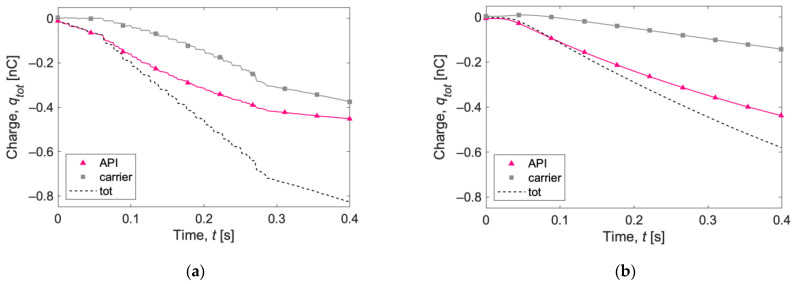
Total net charge acquired by (**a**) 25 mg and (**b**) 1 mg of powder after capsule shaking. The lines show instantaneous evaluation of the charges, and the symbols are only for references.

**Figure 9 pharmaceutics-15-01762-f009:**
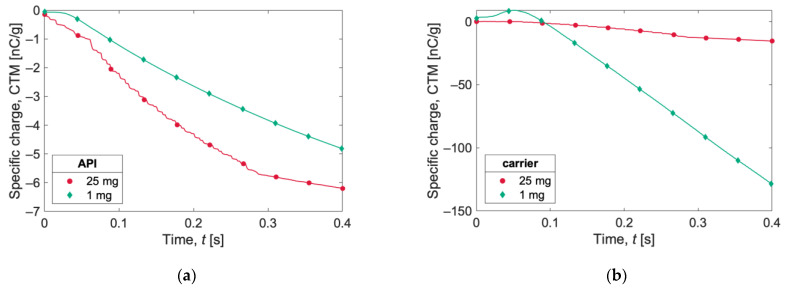
Specific charge expressed as charge-to-mass ratio (CTM) during capsule shaking for (**a**) API and (**b**) carrier particles. The lines show instantaneous evaluation of the charges, and the symbols are only for references.

**Figure 10 pharmaceutics-15-01762-f010:**
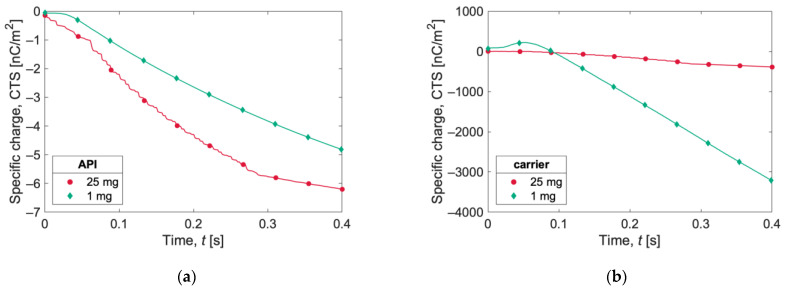
Specific charge expressed as charge-to-surface ratio (CTS) during capsule shaking for (**a**) API and (**b**) carrier particles. The lines show instantaneous evaluation of the charges, and the symbols are only for references.

**Figure 11 pharmaceutics-15-01762-f011:**
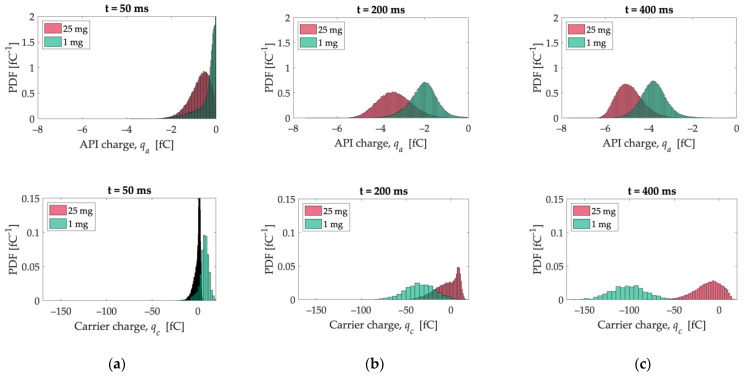
API (**top**) and carrier (**bottom**) charge distribution at different times: (**a**) 50 ms, (**b**) 200 ms, and (**c**) 400 ms.

**Figure 12 pharmaceutics-15-01762-f012:**
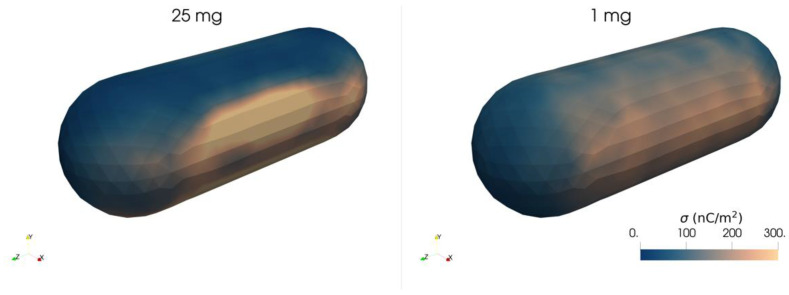
Capsule charge density due to charge exchange with the bottom of the capsule after shaking in the *x*-direction for 60 ms.

**Figure 13 pharmaceutics-15-01762-f013:**
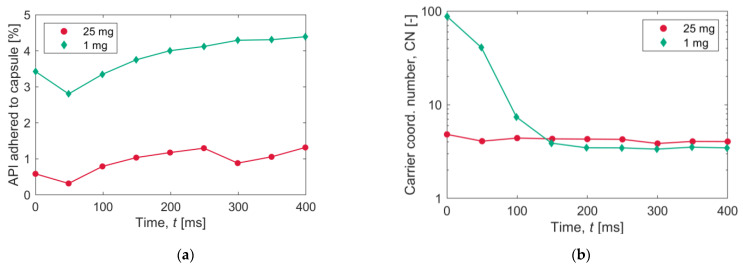
Collision statistics: (**a**) percentage of API particles adhered to the walls and (**b**) mean coordination number (CN) for carrier particles.

**Figure 14 pharmaceutics-15-01762-f014:**
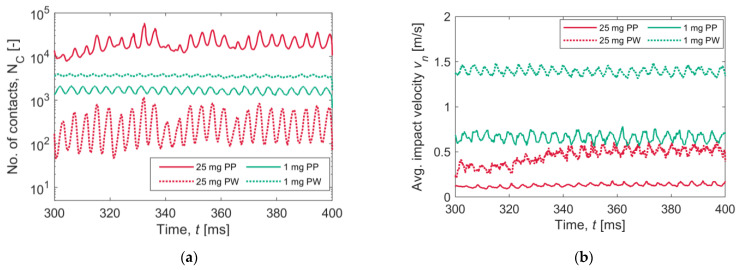
Collision statistics: (**a**) instantaneous number of contacts and (**b**) average impact velocity for the two simulations. Data for particle–particle (PP) and particle–wall (PW) contacts are presented separately.

**Figure 15 pharmaceutics-15-01762-f015:**
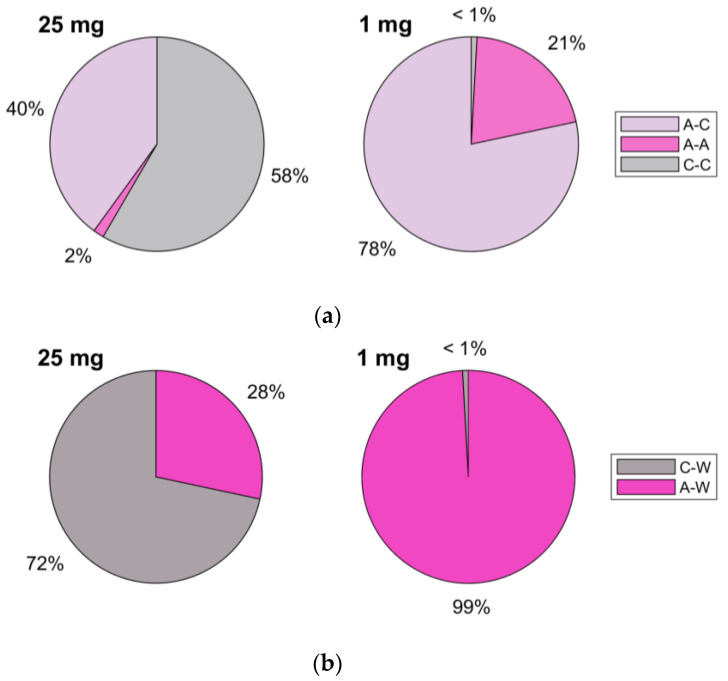
(**a**) Percentage of API–API (A-A), API–carrier (A-C) and carrier–carrier (C-C) collisions with respect to the total particle–particle contacts for the two simulations. (**b**) Percentage of API–wall (A-W) and carrier–wall (C-W) collisions with respect to the total particle–wall contacts for the two simulations.

**Figure 16 pharmaceutics-15-01762-f016:**
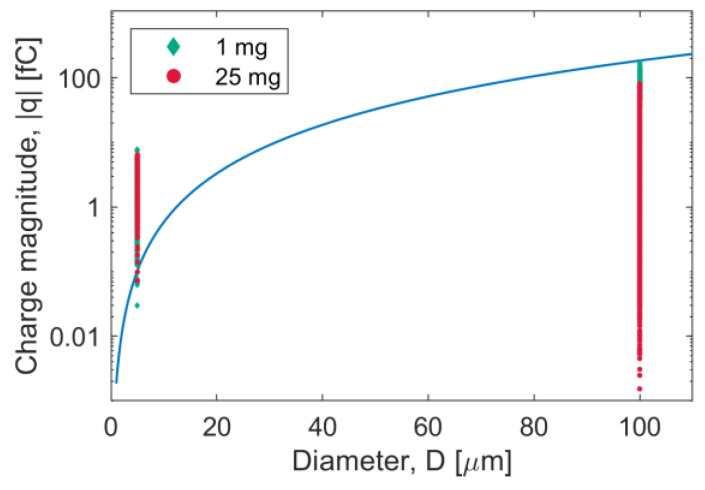
Charge at which P-W electrostatic force is equal to weight as a function of particle diameter (blue line). A scatter plot of the final charge magnitude is reported in the plot for 1 mg and 25 mg simulations, separately for API (5 μm diameter) and carrier particles (100 μm diameter).

**Figure 17 pharmaceutics-15-01762-f017:**
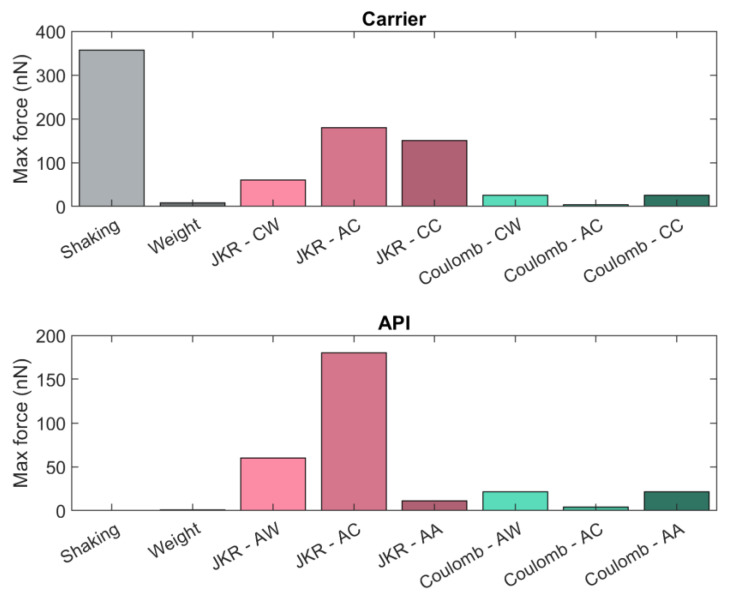
Comparison between estimated maximum force contributions for API and carrier particles. AW = API–wall; CW = carrier–wall; AA = API–API; AC = API–carrier; CC = carrier–carrier.

**Table 1 pharmaceutics-15-01762-t001:** Particles’ mechanical and physical properties.

Property	Carrier	API	Capsule
Reference material	Lactose	Salbutamol	Gelatine
Diameter, *d* (μm)	100	5	-
Density, *ρ* (kg/m^3^)	1500	1200	-
Sliding friction coefficient, *μ_s_* (−)	0.5	0.5	0.5
Rolling friction coefficient, *μ_r_* (−)	0.05	0.05	0.05
Restitution coefficient, *e_n_* (−)	0.85	0.85	0.85
Young modulus, *E* (GPa)	0.2	0.2	0.2
Poisson’s ratio, *ν* (−)	0.35	0.35	0.35
Work function, *Φ* (eV)	5.18	7.70	4.60

**Table 2 pharmaceutics-15-01762-t002:** Surface energy and reference pull-off force values.

	Pull-Off Force, F_pull-off_ (nN)	Surface Energy, *γ* (mJ/m^2^)
API–API	11	0.93
API–carrier	180	8.02
Carrier–carrier	150	0.64
Particle–wall	60	2.55

**Table 3 pharmaceutics-15-01762-t003:** Characteristics of the powder in the two simulated configurations.

	A	B
Sample total mass	25 mg	1 mg
Total no. of particles in sample	123,858	117,233
No. of API particles	92,883	115,825
No. of carrier particles	30,975	1408
API-to-carrier ratio (*w/w*)	1:3332	1:124
API loading (*w/w*)	0.03%	0.80%
Total surface area (cm^2^)	9.80	0.52

**Table 4 pharmaceutics-15-01762-t004:** Charging level after gravity settling.

		Net Charge (pC)	CTM (nC/g)	CTS (nC/m^2^)
API	25 mg	−10.54	−1445	−1445
	1 mg	−4.77	−525	−525
Carrier	25 mg	3.09	0.13	3.17
	1 mg	2.95	2.67	66.67
Sample	25 mg	−7.46	−0.38	−7.60
	1 mg	−1.82	−1.64	−34.18

## Data Availability

The data presented in this study are available from the corresponding authors upon request.

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
