# Peer review of "Discrete Element Method Evaluation of Triboelectric Charging Due to Powder Handling in the Capsule of a DPI"

_pharmaceutics, 2023, doi:10.3390/pharmaceutics15061762_

Round 1

Reviewer 1 Report

As far as I consider the article as very interesting and very important there is one part quite critical I can’t agree with. Validation/verification of the model with experimental data is a critical part of every model and simulation development. In silico or on paper we can deliver an infinite number of models but this is the only stage where we can justify provided methodology and results.

You are using data provided by Chen which is a very nice source to base on but the results of your model are clearly not explaining observations provided by researchers. Change in charge was clearly not linear and your model shows linear behavior in this example. There is something missing and I can’t recognize it as a good fit and explanation because you either state that Chen was wrong (but in that case, experimental proof is needed) or your model has some missing parts important for the process. I admire the amount of work and lots of results you provided but I can’t consider them real without this part to be solved.

Did you try adding to the simulation some smaller particles of lactose and make the simulation for bimodal of 3 modal distribution (d10/d90) – maybe that’s the issue here as far as we can recognize different behavior of smaller and larger particles. Maybe you tried something else and you can provide another explanation? Did you try to modify slightly parameters of the model to check the impact on that simulation and comparison to experimental data?

Looking at other parts of the manuscript I can just ask to look for minor typos – not many of them but we can find them like on page 3: “din” in the last sentence on page 3.

Author Response

Thank you for the comments. Please see the attached file for a detailed reply.

Reviewer 2 Report

The following remarks should be addressed to authors

Subsection 2.1. DEM simulations

Why are the equations not numbered consequently?

Page 3 the last paragraph “JKR, CDT models were implemented din the original version of the code.” – needs correction

Subsection 2.2 In this section numbering of equations starts with 9, after that continues with 2, 3 …Authors need to consider numbering.

The 4th paragraph –“ The first term of the sum in brackets in Equation (2) is the contact potential difference…” - there are two equations numbered (2). Which equation do the authors mean?

2.2 According to the instructions “All Figures, Schemes and Tables should be inserted into the main text close to their first citation and must be numbered following their number of appearance”.  The first mentioned table is “Table 2”. Authors have to correct table numbering in this subsection and in the next one.

Subsection 3.5 – The number of equation has to be considered.

Section 4  The net charge is higher for the highest loaded dose (25 mg), but the specific charge is higher for the lowest loaded dose (1 mg).” -This sentence is unclear. What do the authors mean? Do they mean carrier-rich or API-rich formulations?

Author Response

Thank you for your comments. Reviewer's comments are reported below denoted by "R:" and authors' reply by "A:" .

R: Why are the equations not numbered consequently?

Page 3 the last paragraph “JKR, CDT models were implemented din the original version of the code.” – needs correction

Subsection 2.2 In this section numbering of equations starts with 9, after that continues with 2, 3 …Authors need to consider numbering.

The 4th paragraph –“ The first term of the sum in brackets in Equation (2) is the contact potential difference…” - there are two equations numbered (2). Which equation do the authors mean?

2.2 According to the instructions “All Figures, Schemes and Tables should be inserted into the main text close to their first citation and must be numbered following their number of appearance”.  The first mentioned table is “Table 2”. Authors have to correct table numbering in this subsection and in the next one.

Subsection 3.5 – The number of equation has to be considered.

A:  The numbering of equations and tables has been corrected, thank you for pointing it out.

R: Section 4  “The net charge is higher for the highest loaded dose (25 mg), but the specific charge is higher for the lowest loaded dose (1 mg).” -This sentence is unclear. What do the authors mean? Do they mean carrier-rich or API-rich formulations?

A: The sentence in sentence 4 has been rewritten as follows to improve clarity: “The net charge is higher for the carrier-rich 25 mg formulation, but the specific charge is higher for the API-rich 1 mg formulation.”

Reviewer 3 Report

The research is well-designed and written. The minor concerns are to avoid/explain abbreviations used in the title or abstract.

Some more data regarding this research work must be included in the introduction and their shortcomings should be highlighted.

The conclusion should be comprehensive with the main focus on the novelty of your research work.

Minor changes are required.

Author Response

Thank you for your comments. Please find below reviewer's comments denoted by "R:" and authors' reply denoted by "A:" .

R: The research is well-designed and written. The minor concerns are to avoid/explain abbreviations used in the title or abstract.

A: Thank you. The abbreviations have been expanded or explained both in the title and abstract.

R: Some more data regarding this research work must be included in the introduction and their shortcomings should be highlighted.

A: Shortcomings of several previous works have been added to the introduction. Since the list of cited references in the introduction is already long (39 works), we are not so keen on extending it further with generic works. If the reviewer has specific relevant suggestions, we would be happy to consider them for inclusion.

R: The conclusion should be comprehensive with the main focus on the novelty of your research work.

A: Conclusions have been improved with specific references to the novel aspects of the work.

Round 2

Reviewer 1 Report

Dear Authors,

in light of the presented results and changes introduced in the manuscript, I do recommend your article for publication.